# Feasibility and safety of targeted focal microwave ablation of the index tumor in patients with low to intermediate risk prostate cancer: Results of the FOSTINE trial

Nicolas Barry Delongchamps[1,2]*, Alexandre Schull[1], Julien Anract[1,2], Jean-Paul Abecassis[3], Marc Zerbib[1], Mathilde Sibony[4], Léa Jilet[5], Hendy Abdoul[5], Vincent Goffin[2], Michaël Peyromaure[1,2]

1 Department of Urology, Cochin Hospital, APHP, Paris Descartes University, Paris, France, 2 Inserm Unit U1151, Paris Descartes University, Paris, France, 3 Imagerie Paris-Centre, Paris, France, 4 Department of Pathology, Cochin Hospital, APHP, Paris Descartes University, Paris, France, 5 Clinical Research Unit, Cochin hospital, APHP, Paris Descartes University, Paris, France

* nicolas.barry-delongchamps@aphp.fr

## Abstract

### Objective

To assess the feasibility, safety and precision of organ-based tracking (OBT)-fusion targeted focal microwave ablation (FMA), in patients with low to intermediate risk prostate cancer.

### Patients and method

Ten patients with a visible index tumor of Gleason score ≤3+4, largest diameter <20mm were included. Transrectal OBT-fusion targeted FMA was performed using an 18G needle. Primary endpoint was the evidence of complete overlap of the index tumor by ablation zone necrosis on MRI 7 days after ablation. Urinary and sexual function were assessed with IPSS, IIEF5 and MSHQ-EjD-SF. Oncological outcomes were assessed with PSA at 2 and 6 months, and re-biopsy at 6 months.

### Results

Median [IQR] age was 64.5 [61–72] years and baseline PSA was 5 [4.3–8.1] ng/mL. Seven (70%) and 3 (30%) patients had a low and intermediate risk cancer, respectively. Median largest tumor axis was of 11 [9.0–15.0] mm. Median duration of procedure was of 82 [44–170] min. No patient reported any pain or rectal bleeding, and all 10 patients were discharged the next day. Seven days after ablation, total necrosis of the index tumor on MRI was obtained in eight (80% [95%CI 55%-100%]) patients. One patient was treated with radical prostatectomy. Re-biopsy at 6 months in the other 9 did not show evidence of cancer in 4 patients. IPSS, IIEF-5 and MSHQ-EjD-SF were not statistically different between baseline and 6 months follow up.

**Data Availability Statement:** All relevant data are within the manuscript and its Supporting Information files.

**Funding:** The trial was funded by KOELIS (koelis. com) to the clinical research unit of "Assistance Publique - Hôpitaux de Paris" (sponsor of the study).

**Competing interests:** I have read the journal's policy and the authors of this manuscript have the following competing interests: N Barry Delongchamps is Consultant for Koelis. This does not alter our adherence to PLOS ONE policies on sharing data and materials.

## Conclusions

OBT-fusion targeted FMA was feasible, precise, and safe in patients with low to intermediate risk localized prostate cancer.

## 1. Introduction

Although prostate cancer is frequently a multifocal disease, the index tumor is believed to have the most malignant potential among other smaller lesions (secondary tumors) within the prostate. Tumor volume and Gleason score are predictive of disease recurrence after radical treatments [1, 2]. These observations led to the option of Focal therapy (FT) of the index tumor, aiming to decrease the risk of cancer progression, while preserving genitourinary function. A number of sources of energy have been employed so far, and different sizes of ablation have been proposed, including tumor-only, zonal or hemi-gland ablation [3].

Interest in FT has recently been renewed owing to improved biopsy and imaging techniques, allowing a more comprehensive management of the index tumor. The ability of magnetic resonance imaging (MRI) to detect significant cancer foci [4–9], together with the reliability of organ-based tracking (OBT) MRI-ultrasound fusion [10–12], now allows clinicians to detect and target the index tumor precisely: Performing an elastic fusion between MRI and ultrasound images can create a precise three-dimensional mapping of the prostate, accurately showing the index tumor. OBT enables physicians to guide and to distribute, in a real-time fashion, the biopsy cores in three-dimensions. Another major value of OBT resides in its capacity to memorize and then recall the location of interest in the prostate, from biopsy to FT, and then during follow-up, providing quality control.

Dodd et al. demonstrated microwave energy resulted in coagulation necrosis with a low "heat-sink" effect, while also reporting the volume of necrosis obtained seemed to be predictable and repeatable [13].

We hypothesized that combining the advantages provided by OBT-targeting and microwave therapy would allow a safe and precise FT of index tumor-only. We thus undertook a prospective trial to assess the feasibility, safety and precision of OBT-fusion targeted focal microwave ablation (FMA), in patients with localized PCa of low to intermediate risk of progression.

## 2. Patients and method

This trial was approved by the French national committee for ethics (CPP, ref: Am7730-2-3439) and registered on ClinicalTrials.gov (NCT03023345). All patients gave their written consent before inclusion.

Between September 2017 and 2018, 10 eligible patients were included. The inclusion criteria were: (1) age between 45 and 76 years; (2) prostate cancer with a visible index tumor on MRI, confirmed on targeted biopsy, and (3) signed informed consent. Exclusion criteria were: (1) PSA level >15ng/ml, (2) severe low urinary tract symptoms defined by IPSS>18, (3) index tumor largest diameter >20mm, (4) lower distance between index tumor and rectum wall <5mm, (5) evidence of extra-capsular extension or seminal vesicle invasion on MRI, (6) evidence of Gleason grade 4 on systematic biopsies, (7) more than 50% of Gleason grade 4 on targeted biopsy, (8) contraindication to general anesthesia, and (9) untreated bacteriuria less than 2 days before surgery. No protocol violations were identified.

The primary objective was to demonstrate the feasibility and precision of targeted FMA under OBT registration. The primary endpoint was the evidence of complete overlap of the index tumor by ablation zone necrosis on MRI performed 7 days after ablation.

Secondary endpoints were: (1) intra and postoperative outcomes, including safety and pain evaluation, (2) urinary and sexual outcomes, assessed with IPSS, IIEF5 and MSHQ-EjD-SF self-questionnaires, respectively (3) oncological outcomes, assessed with PSA at 2 and 6 months, as well as prostate re-biopsy, at 6 months.

All patients received a preoperative rectal preparation (enema) and prophylactic antibiotic using oral fluoroquinolones. Procedures were performed under general anesthesia. No urethral catheterization was performed. The ultrasound probe was inserted transrectally and held with a mechanical arm (Steady Pro™, Koelis, Meylan, France). Ultrasound-MRI image fusion was performed with OBT-registration using Trinity™ station (Koelis, Meylan, France) [14]. Microwave thermal ablation was provided by the TATO generator (Biomedical Srl, Firenze, Italy) using a single 18G needle inserted transrectally. Duration and power of microwave application were set according to a pre-clinical predictive ablation chart. Patients were discharged the next day.

Multiparametric prostate MRI (mpMRI), including T2 and diffusion weighted imaging, as well as dynamic contrast-enhanced imaging, was performed 7 days after ablation and at 6 months. The primary endpoint was evaluated by a first radiologist by comparing baseline prostate MRI with day-7 MRI. The extent of necrosis 7 days after ablation was evaluated with T1- contrast enhanced weighted sequences. In the absence of complete necrosis covering of the index tumor, the proportion of necrosis extent was visually estimated, according to the surface leaved untreated on each MRI image. A second radiologist evaluated the primary endpoint, blinded to patient characteristics. Six months after the procedure, three targeted biopsies were performed in the treated zone, as well as 10 to 12-core systematic biopsies.

A phase II study design was used to estimate the number of necessary subjects. We considered that complete necrosis of the index tumor was desirable in 99% of patients. We also considered that if less than 60% of patients showed complete necrosis of the index tumor, the treatment under investigation would be considered inefficient. With a two-tailed test, a risk alpha of 0.1 and beta of 0.1, a total number of 10 patients was considered sufficient.

Continuous data were reported as means (sd) or medians [IQR] (for non-normal data), while categorical data were summarized as counts and percentages. To estimate the proportion of patients with complete necrosis of the index tumor 7 days after ablation, a two-sided 95% confidence interval was calculated. A Wilcoxon signed-rank test was used to compare the data from the questionnaires, with a risk alpha of 0.05. All analyses were conducted using SAS software 9.4.

## 3. Results

Eleven patients were enrolled. One patient did not meet eligibility criteria, and only 10 patients were finally included and treated according to the pre-established protocol (Fig 1). Median [IQR] age was 64 [61–72], median PSA was of 5 [4.3–8.1] ng/mL, and median prostate volume of 50 [40–55] mL. Seven patients had low and 3 patients had intermediate D'Amico risk cancer, respectively. Three patients had multifocal disease, with evidence of non MRI-visible secondary tumor foci of Gleason 3+3 cancer on systematic biopsies. The 7 other patients had no evidence of cancer on systematic biopsies. Index tumor characteristics are reported in Table 1 and Fig 2. Baseline median [IQR] IPSS, IIEF-5, and MSHQ-EjD were of 8.5 [4–13], 18 [13–23], and 10 [7–13], respectively.

### 3.1. Intra and post-operative outcomes

Median duration of the surgical procedure was of 82 [44–170] min, including a median time of 16 [8–23] and 2 [2–6] min for MRI-ultrasound fusion and targeting, and microwave

**FOSTINE trial Flow Diagram**

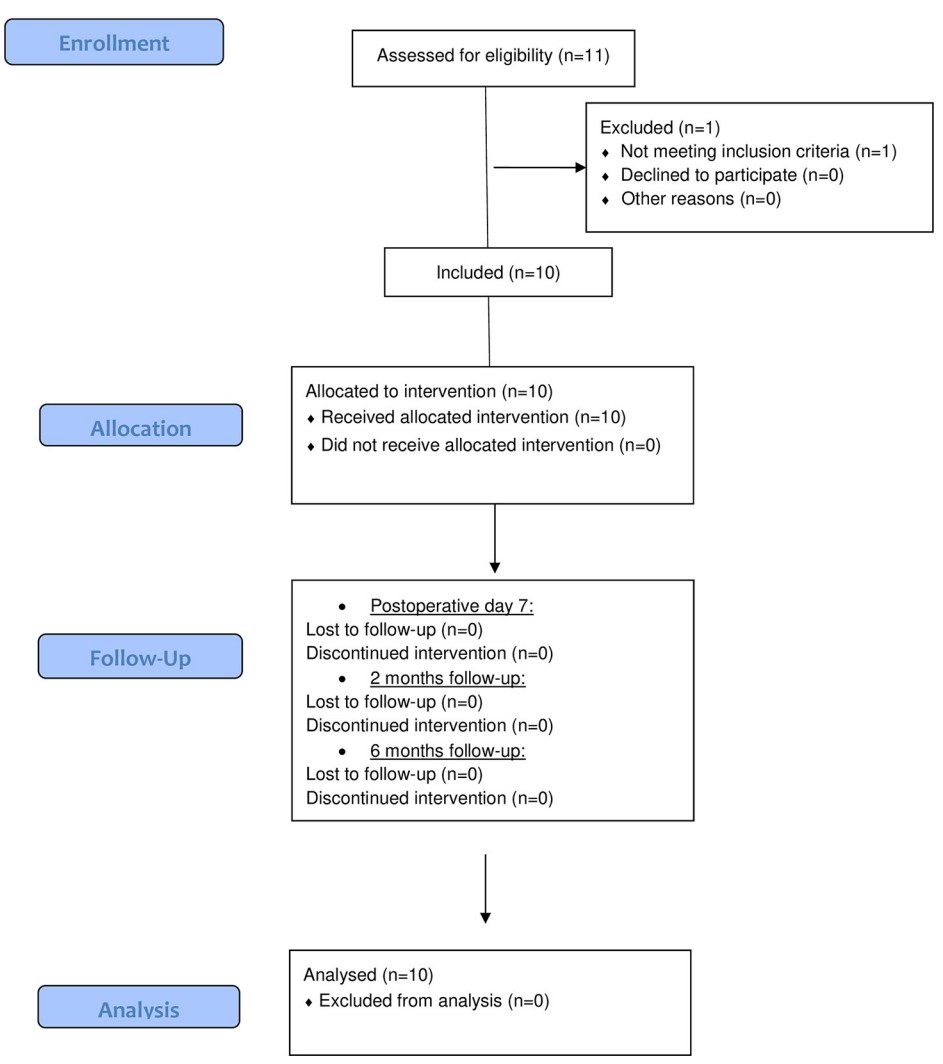

**Fig 1. Flow chart.**

application, respectively (Fig 3). Microwave ablation was set at 15W for 3 min, 15W for 2 min, and 10W for 2 min, in 3 (30%), 5 (50%), and 2 (20%) patients, respectively. In 2 patients, 2 consecutive 3 min applications of 15W were performed.

Postoperatively, all patients recovered spontaneous micturition within the next few hours. No patient reported any pain or rectal bleeding, and all 10 patients were discharged at day 1, according to protocol. None of the patients received any antibiotics after surgery. Outcomes were uneventful, and no adverse events were reported during the 6-month follow-up.

### 3.2. Primary endpoint

Seven days after ablation, total necrosis of the index tumor on MRI was visually confirmed in eight (80% [95%CI 55%-100%]) patients (Fig 4). Median [Q1-Q3] largest dimension of

**Table 1. Summarizes the index tumor characteristics of the 10 patients included.**

| Variable | Median | Interquartile range |
|---|---|---|
| Maximum diameter on mpMRI (mm) | 11.5 | 9.0–15.0 |
| MCCL of targeted biopsies (mm) | 7.0 | 6.0–7.0 |
| Variable | N (total = 10) | % |
| Gleason pattern | | |
| 3+3 | 8 | 80.0 |
| 3+4 | 2 | 20.0 |
| Location | | |
| Apex | 3 | 30.0 |
| Median zone | 3 | 30.0 |
| Base | 4 | 40.0 |

*MCCL: Maximum Cancer Core Length.*

necrosis was of 17.5 [16–22] mm. Table 2 compares, for the 8 patients who had only one microwave application, the largest dimension of necrosis measured on MRI with that of our predictive ablation chart.

In 2 patients (number 5 and 8), necrosis coverage of the index tumor on day 7 MRI was of only 25% and 40%, respectively (Table 3). Patient number 5 had a 20 mm tumor Gleason score 3+3in the anterior base and median zone. Largest diameter of necrosis was 26 mm, obtained with 2 consecutive microwave applications. This patient was exited from the study following unremarkable radical prostatectomy. Surgical specimen analysis showed a pT3a tumor of Gleason 3+3. Patient number 8 had a 15 mm Gleason score 3+3 tumor in the anterior median zone. The largest diameter of necrosis 7 days after ablation was 13 mm. Surveillance was pursued to the 6-month re-biopsy visit (Table 3).

### 3.3. Secondary endpoints

Baseline median total PSA [IQR] was of 5.0 [4.3–8.1] ng/mL. It was of 4.9 [3.7–7.8] and 7.5 [3.9–9.7] ng/mL, at 2 and 6 months, respectively.

Re-biopsy at 6 months was performed in 9 (90%) patients (Table 3). Targeted biopsies of the treated area did not show any evidence of cancer in 4 patients. In the remaining 5 patients, biopsy confirmed persistence of Gleason 3+3 and Gleason 3+4 cancer in 4 and 1 patients, respectively, only in the periphery of the treated area. Systematic biopsies showed Gleason 3+4 cancer outside the treated area in 3 patients (Table 3).

No patient reported any deterioration of urinary or sexual function, but only 6 patients answered to the MHSQ-EjD-SF and 8 patients to the IPSS, IPSS-QoL and IIEF-5 after 6-month follow up. In patients presenting all the data, we did not observe any significant change of median IPSS, IIEF-5, and MSHQ-EjD between baseline and 6-month follow up (Table 4).

### 4. Discussion

As many as seven sources of energy have been tested to ablate pre-defined areas of the prostate [3], Focal HIFU [15] and cryotherapy [16, 17] being the most investigated in terms of number of studies and length of follow-up. Photodynamic therapy is the only focal strategy that has been evaluated in a phase 3 randomized controlled trial [18].

We report here the "first in human" experience of OBT-targeted FMA for the treatment of patients with low to intermediate risk prostate cancer. Our main concern was safety, because

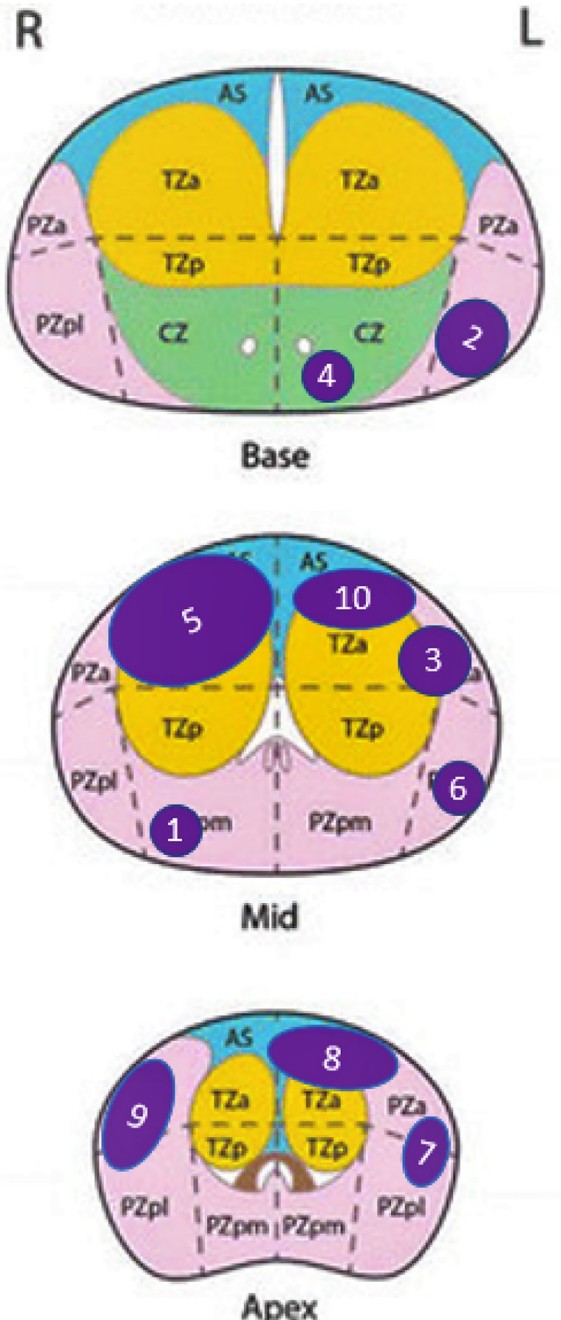

**Fig 2. Index tumor location at baseline in the 10 patients before targeted FMA.** Schematic view of index tumor location. Index tumors have been represented with a 1:1 scale on a prostate with 4 cm height (antero-posterior axis), 6 cm width (lateral axis).

of the proximity of the rectum, especially with the transrectal approach we were using. Our first clinical objective was therefore to confirm the precision of the technique, based on its ability to entirely ablate the target without any harm to immediate adjacent structures.

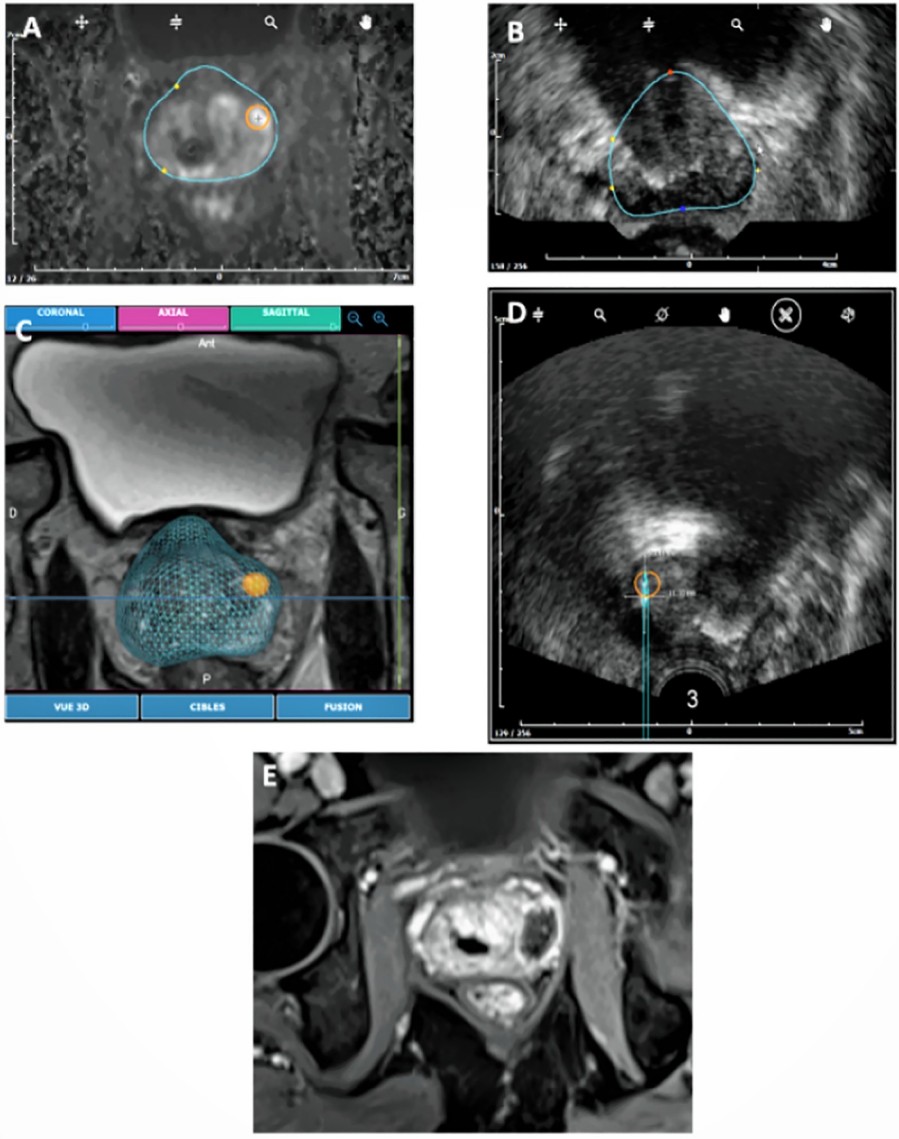

**Fig 3.** Elastic fusion and ablation process in patient N°3: MRI prostate contours were delineated and the target (index tumor) was defined on the station (A). Prostate contours were then delineated on ultrasound images (B), and elastic MRI/Ultrasound fusion was performed (C). This allowed an OBT-Fusion registration of the microwave applicator and measurement on ultrasound of the expected ablation (D), based on ex-vivo predictive ablation charts. The site of ablation was then visualized 7 days after ablation on Dynamic Contrast Enhanced MRI (E).

The procedure was proven to be feasible. OBT-fusion allowed accurate identification of the center of the index tumor. The tip of the microwave applicator was visible under ultrasound and was inserted under visual guidance. The second step consisted in evaluating the range of ablation. We measured the anticipated thermal effect of microwaves from the tip of the applicator, in the 3 planes, and ensured that it was covering all the tumor volume, without any visualization or guidance on what would be the anticipated ablation volume. During the treatment itself, although we did not use real-time monitoring of ablation, the repeated acquisitions with OBT registration allowed us to confirm the adequate position of the microwave applicator. For safety reasons, all patients were treated under general anesthesia, and discharged the next day.

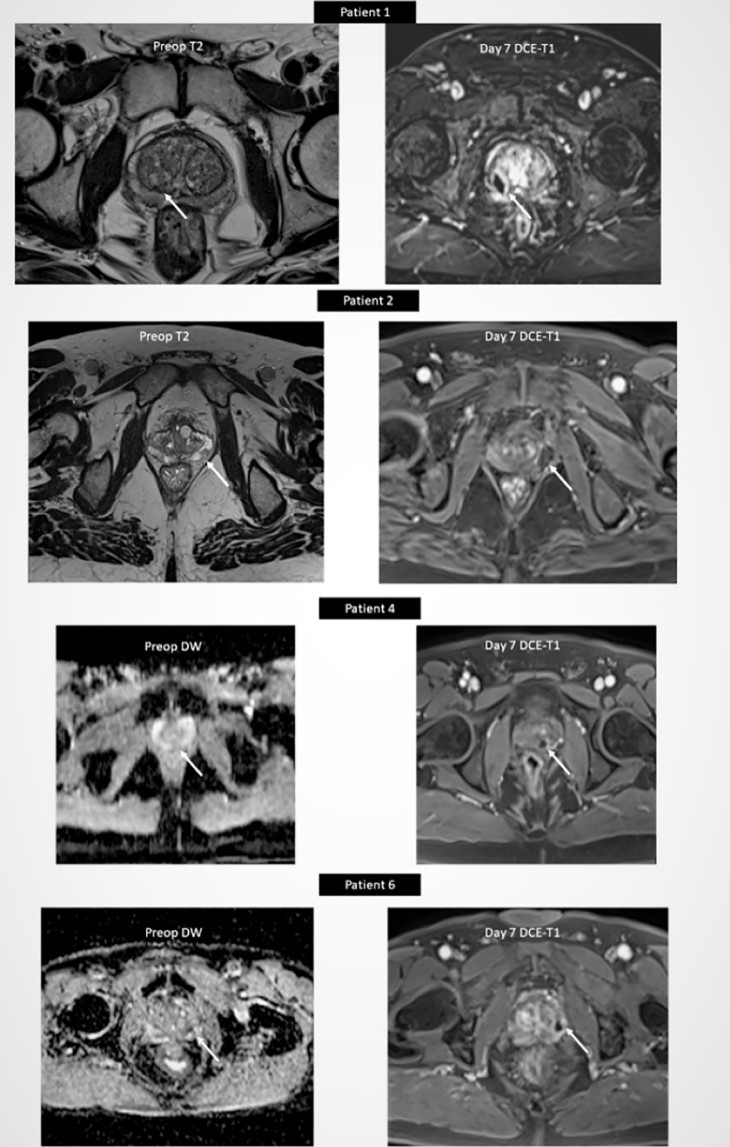

**Fig 4. Assessment of index tumor coverage by necrosis 7 days after ablation in patients 1, 2, 4 and 6: Preoperative MRI the index tumor (arrow).** Postoperative DCE-T1 MRI showing total necrosis of index tumor (arrow).

However, given the absence of pain or any adverse events, we believe this procedure could be proposed under local anesthesia, in outpatient settings.

The results of this study were satisfactory. No patient experienced adverse events nor altered sexual or urinary function during the 6-month follow-up, confirming the safety of this technique. The fact that all patients had immediate micturition after ablation suggests that only minimal local edema was induced by microwave ablation. Also, because the ablation zone of the prostatic necrosis obtained was clearly demarcated on mpMRI, and the volume corresponded to what had been anticipated on pre-clinical predictive ablation charts. Indeed, the largest dimension of the ablation zone measured on the axial view on mpMRI 7days after ablation was within the range of the largest axis of the predictive treated zone for 100% of the 8 patients with only one ablation (Table 2). Last but not least, the OBT-registration allowed us to

**Table 2. Compares the largest dimension of necrosis to that of preclinical ablation charts.**

| N° Patient | Ablation power (in W) | Ablation time (in min) | Largest dimension of ablation on preclinical charts (in mm) | Largest dimension of necrosis on day 7 DCE MRI (in mm) | Difference (in mm) |
|---|---|---|---|---|---|
| 1 | 10 | 2 | 17,00 | 17,74 | 0,74 |
| 2 | 15 | 2 | 20,60 | 18,65 | -1,95 |
| 3 | 15 | 2 | 20,60 | 19,48 | -1,12 |
| 4 | 10 | 2 | 17,00 | 16,50 | -0,50 |
| 6 | 15 | 2 | 20,60 | 19,20 | -1,40 |
| 7 | 15 | 3 | 22,60 | 19,95 | -2,65 |
| 9 | 15 | 2 | 20,60 | 21,10 | 0,50 |
| 10 | 15 | 3 | 22,60 | 21,04 | -1,56 |

**Table 3. Describes the mpMRI and biopsy results of the 10 patients, at baseline, 1 week and 6 months follow-up.**

| Patient Nb | Baseline | | | 1 week | 6 months | |
|---|---|---|---|---|---|---|
| | Index T. Max Diameter (mm) | TB | SB | % necrosis of index T. | TB | SB |
| 1 | 9 | 3+3 | 0 | 100% | 3+3 | 3+3 |
| 2 | 19 | 3+4 (30%) | 0 | 100% | 0 | 0 |
| 3 | 9 | 3+3 | 0 | 100% | 0 | 3+3 |
| 4 | 10 | 3+4 (30%) | 3+3 | 100% | 3+4 (10%) | 3+4 (10%) |
| 5 | 20 | 3+3 | 3+3 | 40% | exited from follow-up (radical prostatectomy) | |
| 6 | 5 | 3+3 | 0 | 100% | 3+3 | 3+4 (10%) |
| 7 | 8 | 3+3 | 3+3 | 100% | 0 | 3+4 (15%) |
| 8 | 15 | 3+3 | 0 | 25% | 3+3 | 0 |
| 9 | 13 | 3+3 | 0 | 100% | 3+3 | 3+3 |
| 10 | 13 | 3+3 | 0 | 100% | 0 | 0 |

TB: targeted biopsies. 3 targeted cores were performed for each biopsy session.

SB: systematic biopsies. 10–12 systematic cores were performed for each biopsy session.

Index tumor maximum diameter and % necrosis were evaluated on mpMRI.

**Table 4. Shows the median scores [IQR] of IPSS, IIEF-5 and MSHQ-EjD-SF at baseline and at 7 days, 2 months and 6 months of follow-up.**

| | Baseline | D7 | M2 | M6 | p-value (baseline vs M6) |
|---|---|---|---|---|---|
| IPSS | 8.5 [4.0–13.0] (n = 10) | 7.0 [2.0–11.0] (n = 10) | 6.0 [3.0–7.0] (n = 9) | 10.0 [8.5–15.0] (n = 8) | 0.55* |
| IPSS-QoL | 1.0 [1.0–2.0] (n = 10) | 1.5 [1.0–2.0] (n = 10) | 0.0 [0.0–1.0] (n = 9) | 2.0 [0.5–3.0] (n = 8) | 0.94* |
| IIEF-5 | 18.5 [13.0–23.0] (n = 10) | 8.0 [3.0–14.0] (n = 9) | 13.5 [2.0–24.5] (n = 8) | 15.0 [4.5–18.0] (n = 8) | 0.39* |
| MSHQ-EjD-SF (function) | 10.5 [7.0–13.0] (n = 10) | 13.5 [9.0–15.0] (n = 8) | 13.0 [11.0–15.0] (n = 7) | 11.5 [9.0–12.0] (n = 6) | 1* |
| MSHQ-EjD-SF (bother) | 0.0 [0.0–1.0] (n = 10) | 0.0 [0.0–0.5] (n = 8) | 0.0 [0.0–1.0] (n = 7) | 1.0 [1.0–2.0] (n = 6) | 0.82* |

*Based on patients presenting all the data (6 patients for MHSQ-EjD-SF and 8 patients for IPSS, IPSS-QoL and IIEF-5).

precisely target the area of interest, thus ensuring proper placement of the microwave applicator. As a result, our primary endpoint was reached, with 80% of patients having their index tumor entirely ablated.

In two patients, we failed to ablate all visible tumor on MRI. The first patient had a large (20 mm) anterior tumor oriented from the base to the median zone. Our transrectal approach was

not adapted to the shape and the volume of this tumor. Although we performed two consecutive ablations, targeted to the base, and then further to the median zone, we failed to cover entirely the tumor. The second patient had a smaller tumor, but with its largest axis oriented transversely. The thermal ablation generated with the applicator has a shorter transverse range, and the transrectal approach only allowed to reach the tumor perpendicular to its longest axis. We thus failed to achieve a full ablation for all of the tumor. These two failures were not related to the energy used, but only to the surgical approach, and the absence of real-time recall and visualization of the anticipated ablation volume. Real-time recall of the ablation volume is of major importance, especially when more than one microwave application is needed. The software we used at the time of this first pilot study did not allow such guidance. The current developments of the technique aim to improve its precision in specific cases such as those described here. Also, the possibility of a transperineal approach, alone or combined with the transrectal one, will allow to personalize ablation to the shape and size of each tumor.

Although this pilot study was not designed nor powered for such evaluation, early oncological outcomes were evaluated with prostate re-biopsy at 6 months. Our results showed insufficient local control at 6 months in 4 additional patients to the 2 initial failures. In these patients, although focal ablation seemed satisfactory on MRI 7 days and 6 months after ablation, re-biopsy suggested insufficient ablation volume. These results raise the issue of treatment margins. Although there is no true consensus on the size of treatment margin to apply, such strategy seems relevant, especially when performing ultrafocal treatments such as laser ablation [19–21] or microwaves. A report from a consensus meeting in 2015 suggested that a circumferential margin of 5 mm around a lesion that was seen on imaging may be enough [22]. Our group [23] and others [24, 25] reported that MRI underestimated pathological volume in up to 50% of the cases, and that this underestimation was higher for small foci [23]. In this pilot study, the primary endpoint was complete tumor ablation (necrosis) based on MRI findings. In a clinical setting, this endpoint should clearly be extended on a case by case basis, depending on the tumor volume itself, and also on tumor index location and proximity to the capsule and rectum. This kind of personalization would be probably more relevant than applying a standardized length of treatment margin systematically to all patients. Another issue is the potential pretherapeutic understaging and/or progression of secondary cancer foci. Kenigsberg et al. [26] recently reported that approximately 20% of candidates who met predefined criteria for focal ablation, and underwent radical prostatectomy were found to have Gleason pattern 4 outside the hypothetical ablation zones. Whatever the efficacy of any ablation that is not radical, and probably even more if this ablation is adopted for focal therapy, urologists will have to monitor closely patients and inform them of the risk of local relapse.

Our study has several limitations, one of them being the absence of a clear histological definition of ablated tissue. Although we did perform 6-month biopsy in the treated zone, and did observe the ablated areas, the fine analysis of structures ablated is still ongoing and was beyond the scope of the present study. Analysis of radical prostatectomy specimen from patients treated with OBT-fusion targeted FMA would definitely confirm the ablative efficacy of this new treatment. In addition, because this study was a first in human, only a small number of patients were included. This is however the very nature of feasibility studies, with a first step evaluation of safety and precision. A multicenter prospective trial is under preparation to further evaluate the oncological efficacy of OBT-fusion targeted FMA in patients with prostate cancer.

## 5. Conclusions

OBT-fusion targeted FMA is feasible, precise and safe in the treatment of the index tumor of patients with low to intermediate risk localized prostate cancer.

## Supporting information

**S1 Checklist.**
(DOCX)

**S1 File. FOSTINE_stat report.**
(DOCX)

**S2 File. Statement from APHP research.**
(PDF)

**S3 File. CIP_FOSTINE_French version without logo.**
(PDF)

**S4 File. CIP_FOSTINE_V3_English version without logo.**
(PDF)

## Acknowledgments

The sponsor of FOSTINE was the Clinical Research Unit from Cochin Hospital, APHP, Paris Descartes University.

## Author Contributions

**Conceptualization:** Nicolas Barry Delongchamps.

**Data curation:** Léa Jilet, Hendy Abdoul.

**Funding acquisition:** Nicolas Barry Delongchamps.

**Investigation:** Nicolas Barry Delongchamps, Alexandre Schull, Julien Anract, Jean-Paul Abecassis, Marc Zerbib, Mathilde Sibony.

**Methodology:** Nicolas Barry Delongchamps, Léa Jilet, Hendy Abdoul.

**Project administration:** Nicolas Barry Delongchamps, Léa Jilet, Hendy Abdoul, Vincent Goffin.

**Resources:** Alexandre Schull, Hendy Abdoul, Michaël Peyromaure.

**Software:** Léa Jilet.

**Supervision:** Nicolas Barry Delongchamps, Vincent Goffin.

**Validation:** Nicolas Barry Delongchamps, Léa Jilet, Hendy Abdoul.

**Visualization:** Vincent Goffin, Michaël Peyromaure.

**Writing – original draft:** Nicolas Barry Delongchamps.

**Writing – review & editing:** Nicolas Barry Delongchamps, Alexandre Schull, Julien Anract, Marc Zerbib, Mathilde Sibony, Hendy Abdoul, Vincent Goffin, Michaël Peyromaure.

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
