## [Decision Letter · Decision Letter 0]

7 Jan 2021

PONE-D-20-31679

Feasibility and safety of targeted focal microwave ablation of the index tumor in patients with low to intermediate risk prostate cancer: results of the FOSTINE trial

PLOS ONE

Dear Dr. Delongchamps:

Thank you for submitting your manuscript to PLOS ONE. After careful consideration, we feel that it has merit but does not fully meet PLOS ONE’s publication criteria as it currently stands. Therefore, we invite you to submit a revised version of the manuscript that addresses the points raised during the review process.

We look forward to receiving your revised manuscript.

Kind regards,

Neal Shore, MD FACS

Academic Editor

PLOS ONE

Journal Requirements:

2. For more information on PLOS ONE's expectations for statistical reporting, please see https://journals.plos.org/plosone/s/submission-guidelines.#loc-statistical-reporting. Please update your Methods and Results sections accordingly.

3.Thank you for stating the following in the Competing Interests section:

"I have read the journal's policy and the authors of this manuscript have the following competing interests:

N Barry Delongchamps is Consultant for Koelis"

Reviewers' comments:

Reviewer's Responses to Questions

**Comments to the Author**

1. Is the manuscript technically sound, and do the data support the conclusions?

Reviewer #1: Partly

Reviewer #2: Yes

Reviewer #3: Yes

2. Has the statistical analysis been performed appropriately and rigorously? 

Reviewer #1: No

Reviewer #2: Yes

Reviewer #3: Yes

3. Have the authors made all data underlying the findings in their manuscript fully available?

Reviewer #1: Yes

Reviewer #2: Yes

Reviewer #3: Yes

4. Is the manuscript presented in an intelligible fashion and written in standard English?

Reviewer #1: Yes

Reviewer #2: Yes

Reviewer #3: No

5. Review Comments to the Author

Reviewer #1: The manuscript entitled ‘Feasibility and safety of targeted focal microwave ablation of the index tumor in patients with low to intermediate risk prostate cancer: results of the FOSTINE trial’ with the aim to assess the feasibility, safety and precision of organ-based tracking (OBT)-fusion targeted focal microwave ablation (FMA), in patients with low to intermediate risk prostate cancer.

Comments

Abstract & Results

Line 36, Line 127, Line 132, Line 177, word IQR to be stated in the text.

Patients and Methods

Sample size calculation

Line 113-117, 1 or 2 tailed test to be stated. Any reason why alpha 0.10 was chosen?

Statistical analysis

Line 118, statistical analysis section and the statistical test(s) to be stated.

Line 119, the sentence ‘Proportions were reported as means (sd) or medians [IQR], when appropriate.’ requires revision.

Results

Baseline characteristics of the patients to be provided.

Line 156, the sentence Median [Q1-Q3] largest dimension of necrosis was of 17.5 [16-22] mm’ requires revision.

Line 160-164, ensure the data are presented in Table 2.

Table 3, the statistical test to be denoted in the table footnote. Decimal point for p value to be standardized. Effect size, 95% CI could be explored/presented. Table alignment could be improved.

Line 191-192, the number of patients at various time period to be clearly denoted/stated in the table footnote.

Data were collected at various time point but statistical analyses were performed between baseline and 6-month. Nonetheless, due to 'small' sample size, the statistical analysis to be treated with cautious.

Figure 1, the time period, n to be stated. The number of patients assessed/evaluated for each measure to be illustrated in the figure or to be stated in the method/results section.

List of references could be improved e.g. spacing of page number.

Reviewer #2: 1. For the three patients with multifocal disease, what is the maximal core length on non-targeted biopsies and what is the number of systematic biopsies involved out of 10-12 cores ? Did these patients have inferior PSA response at 6 months?

2. What is the rationale of using 1 or 2 treatments? Is it for coverage of a larger tumor or larger margin?

3. The authors described the treated area is based on pre-clinical predictive ablation chart. How did the post-treatment largest dimension of necrosis area on T1-contrast sequence compare with the predictive ablation chart in each of the 10 patients? Please show the largest dimension of the necrosis and also the expected dimension.

4. What is the reason that 2 patients had only 25-40% of the necrosis of the index tumor? Was it due to treatment power (inadequate treatment) or accuracy of targeting (necrosis size adequate BUT not overlapping well with tumor area)?

5. For the patient with radical prostatectomy performed showing pT3a Gleason 3+3, what was the size of the residual tumor and also the size of the treated area?

6. Was the radical prostatectomy difficult after microwave treatment?

7. Microwave treatment in liver would lead to shrinkage of treated area. Would the extent of necrosis be under-estimating the actual treated area ?

8. What is the reason that 4 out of 8 patients with 100% necrosis of index tumor be having residual tumor? Would it be underestimation of tumor by MRI +/- lack of margin coverage?

9. In Figure 2, there are 4 tumors that are in peripheral zone. Are they more than 5mm from rectum during treatment? Would close proximity to rectum be the cause of potential undertreatment and local recurrence? Can you add the Patient number to the tumor locations in Figure 2?

10. Can you show the pre and post treatment MRI films of the 4 tumors in peripheral zone?

11. There are 3 cases with pre-op systematic biopsies showing no Gleason 3+4 cancer but post-op 6 month biopsy showing Gleason 3+4. So, the pre-op MRI missed 3 out of 10 clinically significant cancer, which was much higher than the reported figure (around 90% sensitivity). Would this be due to quality of MRI or just a matter of inadequate systematic sampling (10-12 cores) via transrectal biopsy?

12. Line 251: Can you add margin suggestion in consensus statement on focal therapy?

13. What intra-prostatic margin would you suggest for microwave ablation for prostate tumors?

14. Minor amendments:

a. Please change IPSS-QDV to QOL in English.

b. Line 239: recal -> recall

c. Please update reference 2.

Reviewer #3: This manuscript reported that the “first” experiment of OBT-targeted FMA for the treatment of patient with low to intermediate risk prostate cancer. The main concern was safety of this technique in the patients, which is very significative for the focal therapy in the low to intermediate risk prostate cancer patients.

However, there are some points for improvement in this manuscript.

There are many grammatical and linguistic errors throughout the manuscript. The authors should let the entire manuscript be reviewed by a professional English-speaking writer.

Title

Please explain the word “FOSTINE”.

Abstract

Line 33, the word “at day 7” should be changed into “one week after ablation” or “7 days after ablation”, which is more precise.

Line 33, Please adjust the sentence “Secondary endpoints were adverse events, urinary and sexual functional outcomes, assessed with IPSS, IIEF5 and MSHQ-EjD-SF, and oncological outcomes, assessed with PSA at 2 and 6 months, and re-biopsy at 6 month” to make more clear in the presentation.

Introduction

Line 56, Please explain the word “Index tumor”, and give the theory and significances about ablation on the index lesion of prostate cancer.

Line 58 Through you gave the reference about the “organ-based tracking (OBT) MRI-ultrasound fusion”, may be it is more appropriate to provide a brief explanation and explanation about the OBT MRI-ultrasound fusion.

Patients and Method

Line 79 Why the age in the inclusion criteria is between 45 and 76 years?

Line 78-85 Whether the coagulation function, infection status and cardio-pulmonary function in the inclusion and exclusion criteria.

Line 92-93 The expression that “(2) The urinary and sexual outcomes, assessed with IPSS, IIEF5 and MSHQ-EjD-SF, self-questionnaires, respectively” will be more precise.

Line 97-103 Please describe the preoperative preparation, such as bowel preparation and so on.

Please add images of fusion and puncture or ablation process.

Line 108-109 How to calculate the proportion? By volume or by length?

Line 113-117 How to calculate the sample? What formula or software is used?

Line 119 Please detailed description that the data for application of means (sd) or medians [IQR].

Results

Line 127 “The median age was 64 [61-72] years”.

The number should be described consistently, such as “seven and 3” or “7” and so on.

Line 135 and Line 168 Please change the table caption into a complete sentence.

Line 136 Please draw the diagram properly. This table is very terrible, which is confusion.

Line 145-147 How to judge the ablation necrosis cover the tumor mass during the ablation?

Line 149 Please try to explain the cause of spontaneous micturition in the discussion. Does any patient had hematuresis symptoms?

Line 143-151 How to protect the urethra during ablation? Whether the indwelling catheter was needed during the ablation process? Whether to take antibiotics after ablation?

Line 169 Please draw the table properly.

Discussion

Line 219 The first sentence will be more appropriate if changed into “The results of this study was satisfactory”.

Line 219-220 What does mean for the sentence that “First, no patient experienced adverse events nor was altered sexual or urinary function reported during the 6-month follow-up, confirming the safety of this technique.”

Line 234-235 What does mean for the “but oriented from the right to the left.”

Line 246 In term of oncological outcome, dose any changes was detected from image other than PAS changes.

Line 287-289 Why do the index tumor lesions vary in size from the figure? Should they be scaled according to actual size?

6. PLOS authors have the option to publish the peer review history of their article (what does this mean?). If published, this will include your full peer review and any attached files.

Reviewer #1: No

Reviewer #2: No

Reviewer #3: No

---

## [Author Response · Author response to Decision Letter 0]

12 Feb 2021

Dear Reviewers,

Thank you for your time, your comments and suggestions. All of them have been addressed point by point in the attached document labeled "response to reviewers". We made significant changes accordingly. We hope that this revised version will hold your attention.

Sincerely

---

## [Decision Letter · Decision Letter 1]

2 Mar 2021

PONE-D-20-31679R1

Feasibility and safety of targeted focal microwave ablation of the index tumor in patients with low to intermediate risk prostate cancer: results of the FOSTINE trial

PLOS ONE

Dear Dr. Delongchamps,

Thank you for submitting your manuscript to PLOS ONE. The reviewers  are very pleased to accept your revised manuscript if you would be willing to address only a few additional reviewer comments. If you can address these, then I will rapidly review and then we can proceed with publication. Therefore, we invite you to submit a revised version of the manuscript that addresses the points raised during the review process.

We look forward to receiving your revised manuscript.

Kind regards,

Neal Shore, MD FACS

Academic Editor

PLOS ONE

Journal Requirements:

Reviewers' comments:

Reviewer's Responses to Questions

**Comments to the Author**

1. If the authors have adequately addressed your comments raised in a previous round of review and you feel that this manuscript is now acceptable for publication, you may indicate that here to bypass the “Comments to the Author” section, enter your conflict of interest statement in the “Confidential to Editor” section, and submit your "Accept" recommendation.

Reviewer #1: (No Response)

Reviewer #2: All comments have been addressed

Reviewer #3: All comments have been addressed

2. Is the manuscript technically sound, and do the data support the conclusions?

Reviewer #1: Yes

Reviewer #2: Yes

Reviewer #3: Yes

3. Has the statistical analysis been performed appropriately and rigorously? 

Reviewer #1: (No Response)

Reviewer #2: Yes

Reviewer #3: Yes

4. Have the authors made all data underlying the findings in their manuscript fully available?

Reviewer #1: Yes

Reviewer #2: Yes

Reviewer #3: Yes

5. Is the manuscript presented in an intelligible fashion and written in standard English?

Reviewer #1: Yes

Reviewer #2: Yes

Reviewer #3: Yes

6. Review Comments to the Author

Reviewer #1: Minor comment(s)

Line 127 - two-tailed test to be placed in the statement.

Table 4 - at least 2 decimal points for p value to be presented. The reason for different 'n' in the follow up to be described in the results.

For the demographic characteristics, apart from age, were there any other information could be displayed to describe the patients i.e ethnicity etc.

In the abstract, it was mentioned that IPSS, IIEF-5 and MSHQ-EjD-SF were not statistically different at baseline, 7 days, 2 and 6 months but there was no statistical test performed to look into the difference between the time points except baseline and 6 months. This requires revision.

Reviewer #2: Thank you for addressing all the comments. The figures clearly showed the tumor locations and the table showed the ablation dimensions well.

Reviewer #3: Dear Author:

Thank you for revising your paper. The paper was better than last time in the understanding the meaning of the text and the format. However, there are some small nibs for improvement in this manuscript.

1. Where is reference 1 in the text?

2. The table in the paper should be in third line in format, which including the table top line, column line, and bottom line, but without the vertical line. Obviously, the table 1 and table 2 did not conform to this format. Besides, there were two headers in table 1, which is confusion.

3. Which sequences are included in multi-parameter MRI? Please give a detailed explanation in this paper.

4. The author did not answer the question that how to calculate the proportion of necrosis to the index tumor in the mpMRI 7 days after ablation? By volume or by length? Please describe this in the method.

5. Did these ten patients in this pilot clinical trial take antibiotics after ablation? Please add the information to the text.

6. It is pleasure for the results that there was no urethral injury in all the patients. Could you provide the distance between the lesion and the urethra? I think this can offer certain guidance to the clinical application in the future.

7. In regard to the question that whether to take antibiotics after ablation?

7. PLOS authors have the option to publish the peer review history of their article (what does this mean?). If published, this will include your full peer review and any attached files.

Reviewer #1: No

Reviewer #2: No

Reviewer #3: No

---

## [Author Response · Author response to Decision Letter 1]

8 Mar 2021

The authors would like to thank again the reviewers for their comments and suggestions. All points were addressed, and changes were made accordingly in the manuscript. 

Reviewer #1: Minor comment(s)

Line 127 - two-tailed test to be placed in the statement.

We specified that it was a two-tailed test (line 128).

Table 4 - at least 2 decimal points for p value to be presented. The reason for different 'n' in the follow up to be described in the results.

P values were corrected accordingly. We also added a statement explaining the reason for the different numbers of patients included in the analysis (lines 204-207).

For the demographic characteristics, apart from age, were there any other information could be displayed to describe the patients i.e ethnicity etc.

Unfortunately, we did not include any other demographic characteristic in our CRF.

In the abstract, it was mentioned that IPSS, IIEF-5 and MSHQ-EjD-SF were not statistically different at baseline, 7 days, 2 and 6 months but there was no statistical test performed to look into the difference between the time points except baseline and 6 months. This requires revision.

We apologize for this. The abstract was revised accordingly (line 44).

Reviewer #2: Thank you for addressing all the comments. The figures clearly showed the tumor locations and the table showed the ablation dimensions well.

Reviewer #3: Dear Author:

Thank you for revising your paper. The paper was better than last time in the understanding the meaning of the text and the format. However, there are some small nibs for improvement in this manuscript.

1. Where is reference 1 in the text?

Thank you for raising this point. We corrected (line 54).

2. The table in the paper should be in third line in format, which including the table top line, column line, and bottom line, but without the vertical line. Obviously, the table 1 and table 2 did not conform to this format. Besides, there were two headers in table 1, which is confusion.

We corrected the format of the tables

3. Which sequences are included in multi-parameter MRI? Please give a detailed explanation in this paper.

Multi-parametric pMRI includes T2 and diffusion weighted imaging, as well as dynamic contrast-enhanced imaging (interesting especially for the evaluation of necrosis 7 days after ablation). We added a sentence in the manuscript (lines 115 and 116).

4. The author did not answer the question that how to calculate the proportion of necrosis to the index tumor in the mpMRI 7 days after ablation? By volume or by length? Please describe this in the method.

In the absence of complete necrosis covering of the index tumor, the proportion of necrosis extent was visually estimated, according to the surface leaved untreated on each MRI image. Two radiologists estimated the images, blinded to the patients characteristics. We added the description in the method section (line 120-121).

5. Did these ten patients in this pilot clinical trial take antibiotics after ablation? Please add the information to the text.

None of the patients received any antibiotics after surgery. We added the information in the text (line 165 in the result section).

6. It is pleasure for the results that there was no urethral injury in all the patients. Could you provide the distance between the lesion and the urethra? I think this can offer certain guidance to the clinical application in the future.

Unfortunately, the length between the index tumors and urethra was not recorded in our CRF. The prostatic urethra is included in the parenchyma of the gland, and ablation does not impact long-term functional outcomes. The immediate risks are related to hematuria, but usually mild, and spontaneously resolute after a few days. In our study, as mentioned in the result section, no patient reported any pain or hematuria after ablation.

7. In regard to the question that whether to take antibiotics after ablation?

We do not consider that there is a need to take antibiotic after microwave focal ablation. The different protocols of focal ablation using cryotherapy, HIFU, interstitial laser or electroporation do not recommend the use of postoperative antibiotics.

---

## [Editor Report · Decision Letter 2]

10 May 2021

Feasibility and safety of targeted focal microwave ablation of the index tumor in patients with low to intermediate risk prostate cancer: results of the FOSTINE trial

PONE-D-20-31679R2

Dear Dr. Delongchamps,

We’re pleased to inform you that your manuscript has been judged scientifically suitable for publication and will be formally accepted for publication once it meets all outstanding technical requirements.

Kind regards,

Neal Shore, MD FACS

Academic Editor

PLOS ONE

Additional Editor Comments (optional):

Thank you for your diligence in responding to all of the reviewer comments
---

## [Editor Report · Acceptance letter]

5 Jul 2021

PONE-D-20-31679R2 

Feasibility and safety of targeted focal microwave ablation of the index tumor in patients with low to intermediate risk prostate cancer: results of the FOSTINE trial 

Dear Dr. Barry Delongchamps:

I'm pleased to inform you that your manuscript has been deemed suitable for publication in PLOS ONE. Congratulations! Your manuscript is now with our production department. 

Kind regards, 

on behalf of

Dr Neal Shore 

Academic Editor

PLOS ONE